# Effect of Thermal Stress on Thermoregulation, Hematological and Hormonal Characteristics of Caracu Beef Cattle

**DOI:** 10.3390/ani12243473

**Published:** 2022-12-08

**Authors:** Natalya G. Abduch, Bianca V. Pires, Luana L. Souza, Rogerio R. Vicentini, Lenira El Faro Zadra, Breno O. Fragomeni, Rafael M. O. Silva, Fernando Baldi, Claudia C. P. Paz, Nedenia B. Stafuzza

**Affiliations:** 1Centro Avançado de Pesquisa e Desenvolvimento em Bovinos de Corte, Instituto de Zootecnia (IZ), Sertãozinho 14174-000, SP, Brazil; 2Departamento de Genética, Faculdade de Medicina de Ribeirão Preto (FMRP), Universidade de São Paulo (USP), Ribeirão Preto 14049-900, SP, Brazil; 3Departamento de Zootecnia, Faculdade de Ciências Agrárias e Veterinárias (FCAV), Universidade Estadual Paulista Júlio de Mesquita Filho (UNESP), Jaboticabal 14884-900, SP, Brazil; 4Departamento de Zoologia, Universidade Federal de Juiz de Fora (UFJF), Juiz de Fora 36036-900, MG, Brazil; 5Department of Animal Science, University of Connecticut, Storrs, CT 06269, USA; 6Zoetis, 04710-230, São Paulo, SP, Brazil

**Keywords:** adaptability, *Bos taurus*, hemogram, infrared thermography, thermotolerance

## Abstract

**Simple Summary:**

Heat stress impacts animal production, reproduction, health, and welfare, making it profitable to raise animals adapted to the climate. In this context, Caracu (*Bos taurus taurus*) is the large Creole breed adapted to the Brazilian tropical climate. Thus, this study investigated the influence of environmental temperature on thermoregulation (surface temperature obtained by infrared thermography and rectal temperature), hormonal (cortisol, cortisone, and progesterone), and hematological characteristics of Caracu. The results revealed that heat stress changes several physiological characteristics of Caracu cattle, although most of them remained within the regular values observed for taurine Creole breeds. In addition, males and females exhibited differences in their responses to heat stress. The decrease in animal production caused by heat stress is a growing concern. Therefore, the identification and wide use of naturally resistant animals in terminal crossbreeding schemes to explore heterosis and complementarity, and in pure herds, is important to minimize the impact of heat stress on animal production.

**Abstract:**

This study evaluated the influence of environmental temperature on thermoregulation, hormonal, and hematological characteristics in Caracu cattle. Blood samples, hair length, coat and muzzle colors, rectal (RT), and surface temperatures were collected from 48 males and 43 females before (morning) and after sun exposure for eight hours (afternoon). Infrared thermography (IRT) was used to identify superficial temperature that exhibits a high correlation with RT. Hematological parameters, hormone concentrations, RT, and the superficial temperature obtained by IRT that exhibited the highest correlation with RT were evaluated by variance analysis. Regarding IRT, the lower left side of the body (LS) showed the highest correlation with the RT. Interaction between period and sex was observed for LS, cortisol, and eosinophils. Cortisone, progesterone, and RT were influenced by period and sex. Neutrophils and segmented neutrophils were influenced by the period, which showed the highest concentrations after sun exposure. Platelets, leukocytes, lymphocytes, and monocytes were influenced by sex. Heat stress changes several physiological characteristics where males and females exhibited differences in their responses to heat stress. Furthermore, most characteristics evaluated remained within the regular values observed for taurine Creole breeds, showing that Caracu is adapted to tropical climates.

## 1. Introduction

The increase in environmental temperature can directly affect the production and reproduction of cattle since heat stress negatively impacts the physiological, immunological, and performance characteristics of animals, such as feed intake, body weight gain, carcass composition, and reproductive efficiency [1,2]. In this context, beef cattle are among the main affected livestock species due to the extensive nature of production systems adopted in most countries [3].

Although cattle can adapt to different environmental temperatures, the response mechanisms that ensure their survival also affect their performance. The energy that would be directed to production is redirected to the activation of thermoregulatory mechanisms that will act on the animals’ metabolic energy balance, causing changes such as decreased food intake and a decrease in weight gain [4,5].

Heat stress can lead to adaptive changes, resulting in variations observed between animals to thermotolerance. These traits can be quantified by measuring physiological responses such as rectal temperature and surface temperature [6]. Additionally, hematological and hormonal parameters also can be modified by heat stress since they are directly related to mechanisms of heat loss [7]. Factors that affect the caloric elimination capacity from the skin surface, such as coat color, density, hair length, the thickness of the hair fibers, and both hair and skin color, also appear to be related to heat stress adaptiveness providing higher protection against sunlight and higher animal welfare [8,9,10]. Intrinsic factors such as breed, age, sex, and metabolic rate can directly influence the animal’s response to heat stress [11]. Heat stress can also reduce feed intake, ending up in production and health losses [12]. Although zebu breeds (*Bos taurus indicus*) have higher heat tolerance than taurine breeds (*Bos taurus taurus*) [10], taurine breeds can also be selected for tolerance to tropical climatic conditions, as has been observed in Creole breeds [13,14].

The Caracu cattle breed (*Bos taurus taurus*) was formed after uncontrolled crosses between several Portuguese breeds, including Algariviano, Alentejana, Minhota, Arouquesa, Barrosã, and Mirandesa, brought from the Iberian Peninsula to Brazil in 1534 by Portuguese colonizers. The animals were submitted to the adversities of the tropical climate, as well as the scarce feeding and the presence of parasites, contributing to their adaptation to the tropical and subtropical climate [15]. At the beginning of the 20th century, the genetic breeding program of Caracu was created, which established the breed pattern and promoted its development, becoming 1940 one of the most expressive breeds of Brazilian livestock [16]. However, the breed was almost extinct in the 60s due to the high inbreeding and the introduction of Zebu breeds in Brazil. Currently, the Caracu breed has the largest herd among the Brazilian Creole breeds, with a herd estimated at 30,000 animals [17].

The Caracu breed is well-known for its precocity, adaptability, rusticity, and good productive and reproductive performances [15,16,17,18,19,20,21]. Nowadays, Caracu is widely used in crossbreeding, mainly with zebu breeds, producing animals with a high degree of heterosis and good meat quality traits. Cyrillo et al. [22] observed the following means for rump height, rump length, chest circumference, back length, and body length in Caracu: 137.17 ± 4.57 cm, 41.64 ± 4.04 cm, 192.08 ± 12.54 cm, 98.92 ± 7.90 cm, and 147.22 ± 10.62 cm, respectively. The weight of adult males varies from 800 to 1000 kg and females 450 to 650 kg in extensive breeding systems [22].

Given the difficulty in maintaining animal productivity at high levels and promoting their welfare in a tropical climate, knowledge of the response of Creole breeds to thermal stress is important to develop breeding strategies for these populations. Such strategies would enable their use in terminal crossbreeding schemes to explore heterosis and complementarity and in pure herds since the genetic material of Creole breeds can become more productive in their own environments than exotic breeds. Therefore, the aims of this study were to evaluate the use of infrared thermography as a tool for the assessment of animal heat stress and to evaluate the adaptability to heat stress in Caracu animals through thermoregulation, hormonal, hematological, skin, and hair characteristics.

## 2. Materials and Methods

### 2.1. Animals and Data Collection

All the experimental procedures involving animals were approved by the Institutional Animal Care and Use Committee at the Animal Science Institute, Nova Odessa, São Paulo, Brazil (protocol code CEUA Nº. 292-19, October 7th, 2019), following guidelines for animal welfare according to State Law No. 11.977 of the São Paulo state, Brazil.

A total of 104 Caracu steers (43 females with 335 ± 35.15 kg and 61 intact males with 421 ± 45.07 kg), with an average age of 16 months, from the same herd located in Sertãozinho, São Paulo, Brazil (21°17′ S and 48°12′ W) were used in this study. All steers were certified for breed purity by [20]. All animals were raised in grazing conditions using *Brachiaria sp.* forage and supplemented with mineral salt.

The experiment was performed in February 2019, a month characterized by high temperatures in this region of Brazil, in three days when the black globe temperature was higher than 45 °C. The ITEG-500 (Incon Eletronica Ltd., São Paulo, Brazil) data logger was installed at the paddock where the experiment was conducted for the measurement of ambient temperature, black globe temperature, and relative air humidity in the environment. The outdoor wet bulb globe temperature (WBGT) index was used as an indicator of thermal comfort, which is calculated using the following equation [23]:(1)$$WBTG=0.7t_{nwb}+0.2t_{bg}+0.1t_{db}$$
where *t_nwb_* is the natural wet bulb temperature obtained under sun and wind, *t_bg_* is the black globe temperature measured, and *t_db_* is the dry bulb temperature. Thus, the influence of each temperature is weighted 70%, 20%, and 10%, respectively.

In order to minimize the handling-stress, all animals were brought from the pasture to a paddock with water and food (60% sorghum silage, 25% ground corn, 13% soybean meal, 1.75% mineral salt, and 0.2% urea) ad libitum 21 days before the experiment. During this adaptation period, all animals were taken to the corral twice a day (4:00 AM and 2:00 PM) to simulate the experiment that would be performed later.

All animals were tested before and after sun exposure, hereby termed morning and afternoon measurements. In the morning (4:00 to 6:00 AM), all animals were taken to the open corral, where phenotypes (rectal and surface temperatures) and blood were collected. The animals were then taken back to the paddock with water and food, where they remained exposed to the sun without shade. In the afternoon (2:00 to 4:00 PM), all animals were taken to the open corral again for phenotypes and blood collection.

Rectal temperature (RT) was measured by introducing the Animed model 6200.03 digital clinical thermometer (precision of ± 0.1 °C, Incoterm, Porto Alegre, Brazil) into the animal’s rectum up to ± 5 cm for 2 min. The heat tolerance index (*HTI*) was calculated using the following equation [24]:(2)$$HTI=10-\left(RT2-RT1\right)$$
where *RT*2 is the rectal temperature after heat stress, and *RT*1 is the rectal temperature before heat stress. A value closer to 10 indicates a better-adapted animal.

A thermal imaging camera (FLIR TB420X, FLIR Systems Inc., Willsonville, OR, USA) was used for the body surface temperature measurements, which were obtained from four regions chosen to represent a complete view of the body: left side of the body (from the neck to the rump), muzzle, loin, and left eye. Since the total infrared radiation of an object captured through infrared images represents the sum of the emitted, transmitted, and reflected radiation, the loin and the left side of the body were chosen to observe the total amount of heat [25]. The eye was also measured because it is related to internal body temperature, and it is not affected by environment temperature as other peripheral regions [26]. Additionally, the muzzle is related to heat dissipation mechanisms through respiration and was also measured by infrared thermography [25]. The emissivity value was adjusted (0.98 ε), and the camera resolution was calibrated before each period (morning and afternoon) according to local ambient temperature and humidity. A standard distance of 1 m between the animal and the camera was used for image capture.

The Research IR 4 Software (FLIR Systems Inc., Willsonville, OR, USA) was used to calculate minimum, mean, and maximum temperatures for each measured region. After preliminary analysis, the maximum temperature of each region was used in the statistical analyses to reduce environmental interference with the infrared thermography measurements, as described by [27]. On each image obtained, subregions of measurement were traced using the bendable line tool to detect the targeted subregions for temperature determination in a rainbow palette, as performed by [28] (Figure 1). Two subregions were traced on the left side of the body: upper (US) and lower (LS); two subregions were traced on the muzzle: caudal (CM) and middle (MM); three subregions were traced on the loin: left (LL), middle (LM) and right (LR); two subregions were traced in the left eye: lacrimal canal (LC) and eye globe (EG).

The animals also had their coat color classified as cream, yellow, orange, and red, and hair samples were collected from a central area of the shoulder on both sides using a plier and measured using a pachymeter. The muzzle color was also classified as white, black, or pied.

### 2.2. Blood Sample Collections

The blood samples collected from the jugular vein of each animal in both periods (before and after sun exposure) were used to perform steroid hormones (cortisol, cortisone, and progesterone) quantification and the hemogram analysis (erythrocytes, hemoglobin, hematocrit, mean corpuscular volume—MCV, mean corpuscular hemoglobin—MCH, mean corpuscular hemoglobin concentration—MCHC, leukocytes, band neutrophils, segmented neutrophils, neutrophils, eosinophils, lymphocytes, monocytes, and platelets).

For hormone analysis, blood samples were centrifugated (1500× *g*, 15 min, 10 °C), and 200 μL plasma obtained were prepared in 1:1 methanol (Merck, Darmstadt, Germany) and stored at −80 °C. The quantification of circulating hormones was performed using the high-resolution multiple reaction monitoring (MRMHR) analysis in liquid chromatography-tandem mass spectrometry (LC-MS/MS) using a TripleTOF^®^ 5600 (SCIEX, Foster City, CA, USA) equipment as described by [29]. Identification of the steroid species obtained in the LC-MS/MS analysis was performed using PeakView v.2.1 and MultiQuant™ v.3.0.2 (SCIEX, Foster City, CA, USA) software.

### 2.3. Statistical Analysis

The environmental conditions (wet bulb, dry bulb, black globe, and WBGT index) were analyzed using the GLM procedure of SAS v. 9.2 (SAS Inst. Inc., Cary, NC, USA), considering periods before and after sun exposure as a fixed effect.

Pearson correlation coefficients for each sex were estimated by the CORR procedure of SAS v. 9.2 (SAS Inst. Inc., Cary, NC, USA) to examine the relationship between the maximum temperature of each subregion of measurement obtained by infrared thermography (US, LS, LL, LM, LR, CM, MM, LC, and EG) and RT. The subregion that exhibited the highest correlation with RT was included in the analysis of variance as a covariate.

The hair length, coat color, and muzzle color were analyzed by the GENMOD procedure of SAS v. 9.2 (SAS Inst. Inc., Cary, NC, USA), presuming gamma distribution to hair length. The RT and the temperature obtained by infrared thermography that exhibited the highest correlation with RT were used as covariates to evaluate the hair length, coat color, and muzzle color data.

Hematological parameters (erythrocytes, hemoglobin, hematocrit, MCV, MCH, MCHC, leukocytes, band neutrophils, segmented neutrophils, neutrophils, eosinophils, lymphocytes, monocytes, and platelets), hormone concentrations (cortisol, cortisone, and progesterone), RT and the temperature obtained by infrared thermography that exhibited the highest correlation with RT were analyzed using the MIXED procedure of SAS v. 9.2 (SAS Inst. Inc., Cary, NC, USA), and the model included the fixed effects of the period (morning and afternoon) and sex, as well as the interaction between all effects (*p* < 0.05).

## 3. Results

### 3.1. Environmental Conditions and HTI

All the environmental conditions evaluated showed significant differences (*p* < 0.0001) between periods (Table 1), without differences observed among collection days. The afternoon environment exhibited the highest temperatures (wet bulb, dry bulb, and black globe) and WBGT index (Table 1). The black globe temperature was under 25 °C in the morning period and above 45 °C in the afternoon period.

The high differences observed in environmental conditions between both periods influenced the HTI of the animals, which ranged from 7.2 to 9.7 (mean of 8.5 ± 0.49, a median of 8.6, and mode of 8.6). No significant difference was observed between HTI observed in males and females.

### 3.2. Rectal Temperature and Infrared Thermography

The highest and lowest body surface temperatures obtained by infrared thermography in the morning period were identified in LC and CM subregions, respectively, in both males and females (Table 2). In the afternoon period, the LM and MM subregions exhibited the highest and lowest temperatures, respectively, also in both sexes (Table 2).

The temperature of all nine subregions measured by infrared thermography from four different regions of the animal’s body presented a high and significant correlation (*p* < 0.001) with RT (Table 3). Females showed higher correlation values with RT in most subregions evaluated (US, LS, MM, LL, LM, EG, and LC), while males exhibited higher correlation values only in CM and LR subregions. The EG subregion had the lowest correlation with RT in both males (0.740) and females (0.719), while the LS subregion exhibited the highest correlation with RT in both males (0.911) and females (0.898). Among all subregions analyzed, LS was used in the further analysis because it presented the highest correlation with RT.

### 3.3. Temperatures, Hematological and Hormonal Traits

The descriptive statistical analysis of the hematological and hormonal traits, RT and LS, can be seen in Table 4.

The interaction between period and sex (Figure 2) influenced the LS (*p* < 0.0001), cortisol (*p* = 0.0304), and eosinophils (*p* = 0.0495). The highest LS was observed in females after exposure to the sun (39.59 °C), while the lowest LS was observed in females before exposure to the sun (33.73 °C). The highest and lowest cortisol concentration was observed in females before sun exposure (7.61 nmol/L) and in males also before sun exposure (3.57 nmol/L), respectively. The highest eosinophils concentration was observed in females after sun exposure (0.782 g/L) and the lowest in males after sun exposure (0.562 g/L).

The RT, cortisone, progesterone, hemoglobin, hematocrit, and MCV were influenced by sex and period, without interaction between them (Table 5). Males presented higher RT (38.85 °C) than females (38.70 °C), and the RT observed after sun exposure (39.51 °C) was higher than before sun exposure (38.03 °C). Females exhibited higher cortisone concentrations (1.55 nmol/L) than males (1.26 nmol/L), and the samples collected after sun exposure had higher cortisone concentrations (1.47 nmol/L) than those collected before sun exposure (1.34 nmol/L). Regarding progesterone, hemoglobin, hematocrit, and MCV, females presented higher means (1.33 nmol/L, 6.03 nmol/L, 0.31 L/L, and 42.08 fL, respectively) compared to males (0.22 nmol/L, 5.73 nmol/L, 0.29 L/L, and 40.92 fL, respectively). In addition, the highest estimated means of progesterone, hemoglobin, hematocrit, and MCV were observed before sun exposure (0.86 nmol/L, 5.98 nmol/L, 0.30 L/L, and 41.60 fL, respectively) compared to after sun exposure (0.69 nmol/L, 5.78 nmol/L, 0.29 L/L, and 41.40 fL, respectively).

The erythrocytes, MCHC, neutrophils, and segmented neutrophils were influenced only by the period (Table 5). The highest erythrocytes concentration was observed before sun exposure (7.34 T/L) compared to after sun exposure (7.09 T/L). Regarding MCHC, neutrophils, and segmented neutrophils, the highest concentrations were observed after sun exposure (19.74 nmol/L, 3.63 g/L, and 3.59 g/L, respectively) in relation to before sun exposure (19.60 nmol/L, 3.46 g/L, and 3.43 g/L, respectively).

The platelets, band neutrophils, leukocytes, lymphocytes, monocytes, and MCH were influenced only by sex (Table 5). Males had higher concentrations of platelets (569.33 g/L) than females (481.31 g/L), while females exhibited higher concentrations of band neutrophils, leukocytes, lymphocytes, monocytes, and MCH than males (Table 5).

### 3.4. Hair and Skin Characteristics

The hair length ranged from 3 to 16 mm. The results showed that RT (*p* = 0.41) and LS (*p* = 0.92) were not significantly influenced by hair length (Figure 3). In this herd, 7% of the animals were classified as cream, 33% as yellow, 40% as orange, and 20% as red, which exhibited RT means of 39.7 °C, 39.6 °C, 39.5 °C, and 39.4 °C, respectively, not differing from each other (*p* = 0.95). Similar results were observed for LS, where the means were 39.1 °C for cream, 39.5 °C for yellow, 39.4 °C for orange, and 39.4 °C for red, not differing from each other (*p* = 0.69).

Regarding the muzzle color, 82% of the animals were classified as white, 15% as pied, and 3% as black. The RT means did not differ between animals with different muzzle colors (*p* = 0.12), being 39.6 °C for white, 39.4 °C for pied, and 39.3 °C for black. Similar results were also observed for LS, with the following means: 39.4 °C for white, 39.3 °C for pied, and 40.0 °C for black, not differing from each other (*p* = 0.57) (Figure 4).

## 4. Discussion

### 4.1. Environmental Conditions and HTI

The increase in environmental temperature can induce the elevation of body temperature, causing physiological disturbances. The environmental conditions evaluated demonstrated the influence of the environment on the adaptive mechanisms of the Caracu animals, in which differential responses were identified for the morning and afternoon periods. Differences observed between the WBGT index in the morning and afternoon periods indicated thermal comfort in the morning and thermal stress in the afternoon period.

The HTI showed that there is variability for thermotolerance to the tropical climate within this herd, which may allow the selection of heat-tolerant animals. Environmental differences during RT measurements directly impact the HTI obtained, which makes it difficult to compare our results with the literature. This fact can be corroborated by [18], which found an HTI mean of 9.7 studying the same Caracu herd used in the present study. The divergence of results is due to the methodology proposed since the RT measurements in the morning were performed by [18] in the presence of sunlight, while in the present study, RT measurements in the morning were performed in the absence of sunlight.

### 4.2. Rectal Temperature and Infrared Thermography

Several tools have been arising as alternatives to assess the impact of environmental factors on heat stress in animals, such as infrared thermography. This tool contributes to animal welfare because it enables the detection of heat stress without contact with the animal, avoiding handling stress [30]. Moreover, thermography enables the measurement of the temperature radiated from different parts of the animal body, increasing the range of possibilities and facilitating data collection management [31].

In our study, all nine subregions evaluated exhibited positive, high, and significant correlations with RT. The US and LS body subregions exhibited the highest correlation with RT among the infrared measurements in both sex and periods evaluated, which could be attributed to the influence of ruminal activity. In cattle, the rumen occupies a large area of the left half of the abdominal cavity. Moreover, heat is one of the products of ruminal fermentation, and the temperature of this region is closely related to the internal core body temperature [32,33]. In addition, several factors can influence infrared thermography, such as environmental conditions, meteorological variables (e.g., wind speed and solar loading), refractive emissivity, the distance between the equipment and the animal evaluated, and breed characteristics [25].

For the muzzle region (CM and MM), lower temperature values are usually recorded. However, the vast blood irrigation in the muzzle contributes to the relationship with internal body temperature [25]. The loin subregions (LL, LM, and LR) were in direct contact with sunlight and more susceptible to changes in weather. Thus, the correlations between loin subregions and RT may be related to thermoregulatory mechanisms in search of homeostasis [34].

It has been suggested that the eye is not affected by ambient temperature and has been considered a useful indicator of core body temperature [35], which can be used for detecting temperament [36], pain [37], or response to stress in cattle [38]. Studies with several cattle breeds reported that the eye had high correlations with RT [9,26]. The LC subregion is more innervated, has more superficial capillaries beds and could be related to the internal core body temperature [27,35].

Although infrared thermography has been used in several studies encompassing many breeds, there are some limitations of this technology, and many factors can influence the thermal readings by the infrared device (*e.g.,* anatomical and physiological characteristics of each breed, environmental conditions, as well as differences in the measurement methodology used) [25,28,30]. Differences observed in correlation values between sex could be a result of those physiological and anatomical variables. The internal temperature of cows can vary during the stages of the estrous cycle [12,28]. However, the animals’ reproductive aspects were not evaluated, which may have affected part of our results. Additionally, there was a difference in size between the animals. Males had greater body volume than females. It is known that surface areas rather than body volumes limit metabolic activity such as thermoregulation [39]. Thereby, females may have greater heat dissipation and consequently higher temperatures captured by thermography, reflecting also higher correlation values.

### 4.3. Temperatures, Hematological and Hormonal Traits

The sex and period interaction modified the LS, where the highest and lowest estimated mean was identified in females after sun exposure and before sun exposure, respectively (Figure 2). Although infrared thermography has been used in several studies encompassing many breeds, there are some limitations of this technology, and many factors can influence the thermal readings by infrared devices, as described previously [25,28,30]. Differences observed between sex could be a result of physiological and anatomical particularities of each sex. In addition, the total infrared radiation captured through infrared thermographic images represents the sum of the emitted, transmitted, and reflected radiation [40]. It also can explain the higher correlation observed between LS and RT in relation to other subregions located in the upper area of the animal’s body, such as US, LL, LR, and LM, which are affected by a higher incidence of solar radiation and, therefore, exhibited higher temperatures than RT and, consequently, lower correlations with RT.

Caracu females presented higher LS temperatures after sun exposure compared to males. Authors reported a negative correlation between body surface area and body weight, where animals with lower weight have higher body surface per unit of weight compared to animals with higher weight [41]. It is known that surface areas rather than body volumes limit metabolic activity such as thermoregulation [41]. In the present study, females had lower body weight (335 ± 35.15 kg) than males (421 ± 45.07 kg), which may explain the higher LS means observed in females after sun exposure when compared to males in the same period since females have higher body surface per unit of weight. Thereby, females may have greater heat dissipation and, consequently, higher temperatures captured by thermography, resulting in higher correlation values.

A significant interaction between sex and period was also detected for serum cortisol concentration (Figure 2), where the highest cortisol level was identified in females before sun exposure, and the smallest cortisol level were identified in males also in the morning period. Although several researchers have been studying the cortisol hormone as a biomarker for heat stress, cortisol levels can be influenced by several factors such as environment, management, adaptation period, and temperament of the animal, making it difficult to use as an indicator of heat stress [42].

The highest cortisol levels in females, when compared to males, was also observed in cattle [43] and goats [44], although some authors have not found differences in cortisol levels between sex in cattle [45,46]. Factors such as breed, handling, behavioral patterns, body condition, and gonadal steroid metabolism can also influence cortisol levels [47,48]. Furthermore, the concentration of plasma cortisol is directly related to the animal’s reactivity [47]. These facts could help to explain the highest cortisol levels observed in females in the morning period, once a higher reactivity was observed in females during all experiments and in our previous study [21].

In this study, no significant differences in cortisol levels were observed between the morning and afternoon periods. In the absence of stressful stimulation, corticotrophin-releasing hormone (CRH) and vasopressin (VP) are released in greater amounts in the early morning, causing an increase in cortisol levels in this period when compared to the afternoon [47]. Differences observed in the literature can be explained by differences in the methodology applied for data collection, as well as breed characteristics [49,50]. Furthermore, cortisol can remain in the animal’s body for 80 to 120 min [51], so the cortisol levels can remain altered for up to two hours after the stressor action.

A significant interaction between sex and period was also detected for eosinophils concentration, where females, after sun exposure, exhibited higher eosinophils concentrations while males presented lower concentrations of eosinophils in the same period. Eosinophils are white blood cells that are analyzed together with other immune system cells to identify systemic stress and acute or chronic inflammatory processes. In our study, heat stress changed the eosinophil concentration, demonstrating a change in immunological patterns, but it was not harmful to the animals’ health since the values observed are within those found in healthy animals. According to [52], calves under heat environments presented a significant decrease in eosinophil concentration. Although Caracu females presented higher concentrations of eosinophils than males, there is no evidence in the literature of a significant difference in eosinophils concentrations between sexes.

The RT is one of the most widely used physiological parameters to measure heat stress in cattle because it indicates the internal temperature of the animal more accurately than other parameters studied [6]. The RT measured after sun exposure (39.51 °C) was higher than obtained before sun exposure (38.04 °C) in Caracu cattle, and males (38.85 °C) showed higher RT than females (38.70 °C). In our study, all animals maintained their RT within the expected temperatures for healthy animals (37.1 to 41.0 °C), which may indicate adaptation to the tropical environment.

Changes in serum cortisone levels act to suppress productive and reproductive functions to support animal maintenance and survival [53]. Cortisone levels were higher in animals after sun exposure than before sun exposure, and higher cortisone levels were identified in Caracu females when compared to males. This result can be explained by the tissue’s ability to increase or decrease cortisol concentration through 11ß-hydroxysteroid dehydrogenase (11ßHSD) enzymes, which convert cortisol into hormonally inactive cortisone [54]. Thus, the increase in cortisone levels after sun exposure and the consequent decrease in cortisol levels in the same period was observed in the present study. Although cortisone is not widely used as a heat stress marker, previous studies indicated that cortisone could be a useful biomarker for this purpose [55,56]. Therefore, additional studies to validate this glucocorticoid as a marker of heat stress are needed.

The highest means of plasmatic progesterone levels was observed in animals before sun exposure when compared to after sun exposure. Regarding sex, females exhibited higher progesterone levels when compared to males. It was expected because it is known that females produce higher concentrations of this hormone [57]. The heat stress effect on plasmatic progesterone levels is controversial. Studies have reported that the progesterone concentration is increased during heat stress [12,58], decreased [59], or did not change [60] in cattle. These differences observed in the literature probably is due to the methodology used or the variation in other factors that influence blood progesterone concentrations, such as the kind of stress (acute or chronic) [53]. Furthermore, progesterone concentrations are determined by differences between the production and the hepatic metabolism, and both are affected by changes in dry matter intake [53], which can explain the highest means of plasmatic progesterone levels observed in Caracu before sun exposure when compared to after sun exposure. Furthermore, glucocorticoids act in several tissues, including the reproductive system, provoking a decrease in hormone production, including progesterone [61]. Therefore, the alteration in the concentration of glucocorticoids due to heat stress observed in this study may have affected the concentration of progesterone.

Clinical and hematological parameters can be used to assess animal welfare [62]. Hematocrit indicates the percentage of erythrocytes in the blood, which are cells rich in hemoglobin, a metalloprotein responsible for transporting oxygen to the body’s tissues. According to [52], animals lose metabolic intermediates in the blood when exposed to chronic heat stress, resulting in decreased concentration of hemoglobin and hematocrit. In the present study, the hemoglobin concentration and hematocrit percentage in Caracu were higher in animals before sun exposure when compared to after sun exposure, and females had the highest averages when compared to males. These results showed that sun exposure could negatively influence the hemoglobin and hematocrit, which were also observed by [50]. According to [62], cattle under prolonged thermal stress exhibit a reduction in the percentage of hematocrit, while the increase in this parameter in animals under shade (absence of thermal stress) is directly related to erythrocytes increase [63].

Erythrocytes are responsible for tissue oxygenation, so the greater the number of erythrocytes directly influences the tissue oxygenation capacity [62]. The concentration of erythrocytes before exposure to the sun was higher than after exposure to the sun, suggesting that exposure to heat stress can negatively influence the concentration of erythrocytes.

Band neutrophil are immature neutrophils usually found in the blood when there is an immune response in the acute phase, while segmented neutrophils are the most mature neutrophilic granulocyte cells abundant in the circulating blood of healthy organisms [64]. In the present study, the concentration of band neutrophils was influenced only by sex, and females had higher concentrations than males. It was observed that the concentration of neutrophils increased in animals after sun exposure when compared to before sun exposure, as well as segmented neutrophils also increased in animals after sun exposure when compared to before sun exposure. According to [65], the concentration of neutrophils is negatively correlated with the serum cortisol concentration, demonstrating that glucocorticoids produced under stress can provoke a reduction in neutrophils. In the present study, a higher concentration of cortisol in animals was also observed in Caracu before sun exposure, and consequently, a lower concentration of neutrophils was observed in the same period. Furthermore, the increase in the neutrophils and segmented neutrophils after sun exposure could be an organism response, which may have interpreted the heat exposure as an inflammatory response [66]. According to [50], glucocorticosteroids released due to heat stress often results in neutrophilia. In addition, the concentrations of neutrophils and segmented neutrophils observed in Caracu, as well as remaining erythrocytes concentrations, keep within the regular values, indicating that this herd is tolerant to heat stress under tropical conditions.

Platelets are blood components that play an important role in blood coagulation [64]. As observed in Caracu, the sun exposure does not affect the concentration of platelets, which was influenced only by sex, with males showing higher concentrations than females. The lower concentration of platelets observed in females can be explained by hormonal influence, as progesterone and estrogen act on the endocrine system, increasing the blood volume and resulting in hemodilution [67]. In addition, testosterone and its metabolites, which are normally found in higher concentrations in males, are important stimulators of hematopoiesis. The number of platelets may be raised by androgens, and there may be an effect on platelet function with increased expression of the thromboxane-A2 receptor, which can promote platelet aggregation [68].

The concentrations of leukocytes, lymphocytes, and monocytes were influenced by sex in Caracu, with females showing higher concentrations than males. Studies have reported metabolic disorders and chronic and infectious diseases in cattle related to changes in leukocyte [69,70], lymphocyte [69,71], and monocyte [51,69] concentrations. However, only healthy animals were used in our study, without the detection of pathologies or infectious diseases. Thus, the higher concentrations of all these cells observed in Caracu females can be attributed as a consequence of the higher reactivity observed in females throughout the experiment and in our previous study [21].

Due to few thermal stress studies performed with Creole cattle breeds, which are already considered adapted to the tropical climate, it was difficult to find studies to corroborate our findings. Furthermore, the diversity of methodologies applied both to subject the animals to heat stress and to collect the phenotypes may be responsible for the discrepancy observed in relation to the hematological and hormonal parameters observed in the literature.

### 4.4. Hair and Skin Characteristics

Although several studies have been reported that animals with dark hair color had higher RT when exposed to high temperatures compared to animals with light hair color [6,72,73], our results showed that hair length, coat color, and muzzle color did not influence the RT and LS traits (*p* > 0.05). Nicolau et al. [74] also not observe the correlation between RT and hair length, hair, or skin colors in Caracu cattle. As suggested by [75], adaptation to hot environments leads to the animal’s resistance to heat stress and not the type of coat. In addition, there is a thermal gradient from the inside to the outside of the animal’s body [76], where the internal temperature is higher and decreases to its periphery (skin and hair), which may explain why RT and LS were not influenced by hair and skin traits evaluated. Thus, our results suggest that hair and skin traits measured herein are not fundamental for heat stress evaluation in Caracu cattle which can be attributed to natural selection for adaptability to the tropical climate that the Caracu breed has been undergoing since its formation.

## 5. Conclusions

Infrared thermography displayed an effective tool to detect the internal temperature of animals according to the methodology proposed in this study, and its use could reduce the stress caused by handling and contributing to animal welfare. In addition, although several thermoregulations, hematological and hormonal characteristics were altered when the animals were under heat stress, most of them remained within the regular values observed for taurine cattle breeds adapted to the tropical climate, evidencing the phenotypic plasticity and adaptability of Caracu to the tropical environment. The climatic changes expected in the coming decades may influence the type of cattle raised in herds around the world under conditions of extreme heat. For this reason, in addition to contributing to the breed’s preservation, it is essential to investigate the thermoregulation physiological aspects of a Creole cattle breed.

## Figures and Tables

**Figure 1 animals-12-03473-f001:**
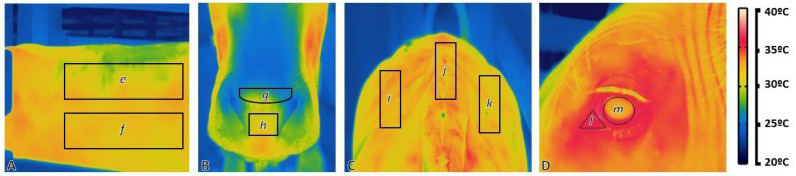
Thermographic images and subregions traced the left side of the body, muzzle, loin, and left eye in Caracu. (**A**) The left side of the body: e = upper (US), f = lower (LS); (**B**) Muzzle: g = caudal area of the muzzle (CM), h = middle of the muzzle (MM); (**C**) Loin: i = left (LL), j = middle (LM), k = right (LR); (**D**) Left eye: l = lacrimal canal region (LC), m = eye globe (EG). Rainbow palette: light colors indicate warmer temperatures (white) and, dark colors indicate cooler temperatures (dark blue), other colors indicate intermediate temperatures.

**Figure 2 animals-12-03473-f002:**
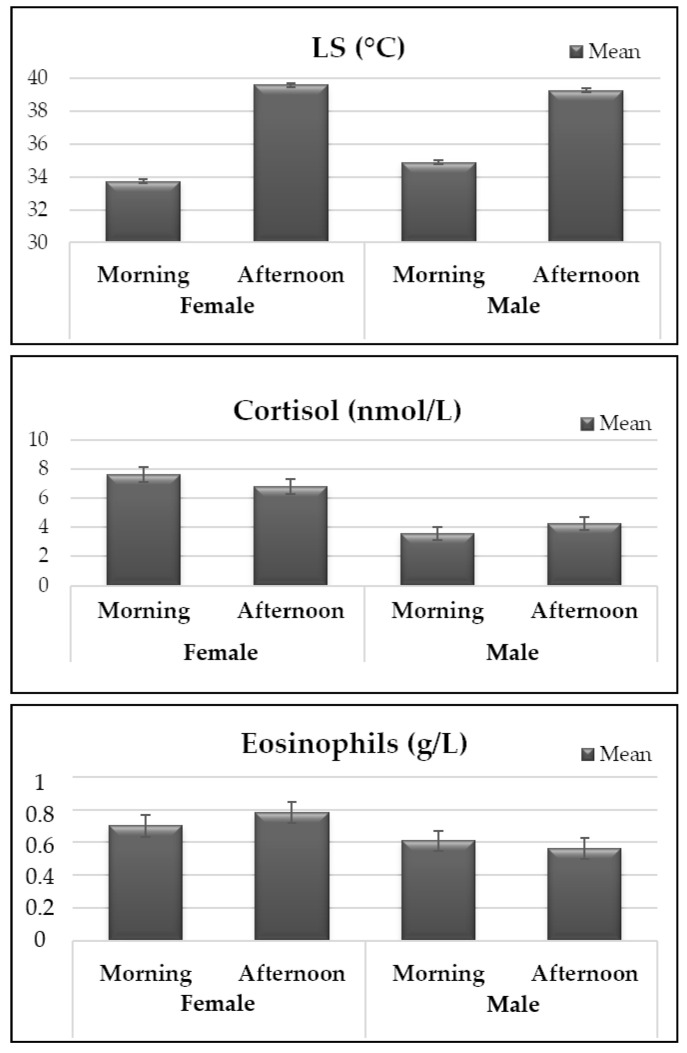
The least square means obtained for the superficial temperature of the lower side of the body (LS) obtained by infrared thermography (*p* < 0.0001), cortisol (*p* < 0.05), and eosinophils (*p* < 0.05) according to sex and period interaction.

**Figure 3 animals-12-03473-f003:**
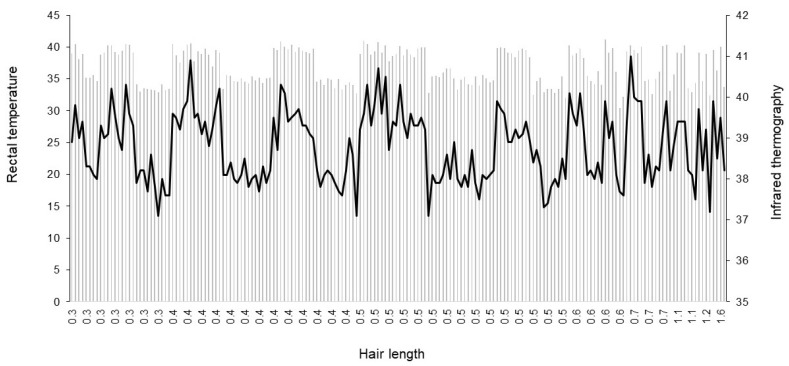
Hair length distribution according to rectal temperature (RT) and the superficial temperature of the lower side of the body (LS) obtained by infrared thermography.

**Figure 4 animals-12-03473-f004:**
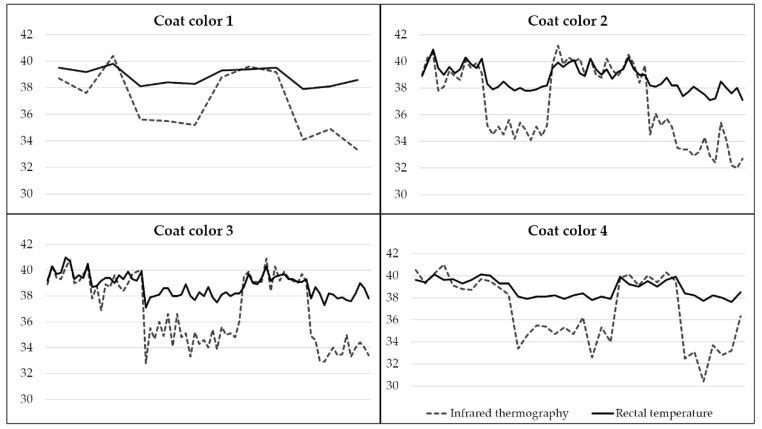
Coat color distribution according to rectal temperature (RT) and the superficial temperature of the lower side of the body (LS) obtained by infrared thermography. Coat color 1: cream; Coat color 2: yellow; Coat color 3: orange; and Coat color 4: red.

**Table 1 animals-12-03473-t001:** Least squares means and standard errors (SE) from environmental conditions obtained at the paddock before (morning) and after (afternoon) sun exposure (*p* < 0.0001).

Temperature (°C)	Morning ± SE	Afternoon ± SE	Variation ± SE
Wet bulb	20.71 ± 0.10	36.35 ± 0.10	15.64 ± 0.14
Dry bulb	20.60 ± 0.09	35.93 ± 0.09	15.33 ± 0.12
Black globe	20.70 ± 0.21	49.15 ± 0.21	28.45 ± 0.30
WBGT index	20.69 ± 0.12	38.87 ± 0.12	18.18 ± 0.16

**Table 2 animals-12-03473-t002:** Mean and standard deviation of rectal temperature and body surface temperature measurements obtained by infrared thermography.

Subregion	Male		Female	
	Morning	Afternoon	Morning	Afternoon
RT	38.09 ± 0.30	39.62 ± 0.51	37.99 ± 0.45	39.40 ± 0.41
US	34.76 ± 0.77	40.45 ± 1.22	33.64 ± 1.29	40.63 ± 0.92
LS	34.89 ± 0.86	39.28 ± 0.88	33.73 ± 1.14	39.59 ± 0.62
MM	34.05 ± 1.00	36.78 ± 0.79	31.77 ± 1.85	36.61 ± 0.68
CM	33.85 ± 1.52	39.06 ± 0.63	30.13 ± 2.75	39.04 ± 0.45
LL	34.75 ± 1.01	40.17 ± 1.21	33.40 ± 0.88	40.36 ± 1.06
LM	34.69 ± 1.04	41.66 ± 1.97	33.47 ± 1.52	41.32 ± 1.10
LR	34.53 ± 0.94	40.16 ± 1.21	33.39 ± 1.17	40.17 ± 1.17
EG	37.36 ± 0.51	38.74 ± 0.98	35.97 ± 1.44	38.47 ± 0.94
LC	38.25 ± 0.51	39.88 ± 0.96	37.60 ± 0.94	39.70 ± 0.50

RT: rectal temperature, US: upper left side of the body US, LS: lower left side of the body, MM: middle of the muzzle, CM: caudal area of the muzzle, LL: loin left, LM: loin middle, LR: loin right, EG: eye globe, LC: lacrimal canal region.

**Table 3 animals-12-03473-t003:** Pearson correlation coefficients (*p* < 0.0001) estimated between the maximum temperature of each subregion of body surface obtained by infrared thermography and rectal temperature (RT) in males (top of the table) and females (bottom of the table).

Subregion	RT	Body		Muzzle		Loin			Eye	
		US	LS	MM	CM	LL	LM	LR	EG	LC
RT	1	0.893	0.911	0.813	0.853	0.889	0.870	0.863	0.740	0.874
US	0.861	1	0.968	0.854	0.913	0.966	0.950	0.958	0.708	0.879
LS	0.898	0.976	1	0.836	0.916	0.949	0.922	0.941	0.726	0.890
MM	0.813	0.877	0.868	1	0.904	0.828	0.843	0.821	0.705	0.838
CM	0.867	0.912	0.923	0.917	1	0.898	0.889	0.895	0.670	0.841
LL	0.872	0.975	0.965	0.851	0.903	1	0.957	0.970	0.707	0.839
LM	0.853	0.963	0.946	0.889	0.911	0.956	1	0.947	0.707	0.842
LR	0.877	0.965	0.963	0.863	0.909	0.962	0.938	1	0.695	0.835
EG	0.719	0.765	0.770	0.709	0.753	0.729	0.707	0.774	1	0.777
LC	0.775	0.798	0.815	0.685	0.737	0.814	0.787	0.819	0.586	1

US: upper left side of the body, LS: lower left side of the body, MM: middle of the muzzle, CM: caudal area of the muzzle, LL: loin left, LM: loin middle, LR: loin right, EG: eye globe, LC: lacrimal canal region.

**Table 4 animals-12-03473-t004:** Descriptive statistics (N: number of observations, mean, SD: standard deviation, SE: standard error, CV: coefficient of variation, minimum and maximum) of the physiological, hormonal and hematological traits observed.

Trait	N	Mean	SD	SE	CV	Min.	Max.
RT (°C)	182	38.78	0.86	0.063	2.21	37.10	41.00
LS (°C)	182	36.88	2.73	0.202	7.41	30.40	41.20
HTI	182	8.52	0.49	0.04	5.75	7.20	9.70
Cortisone (nmol/L)	179	1.40	0.48	0.036	33.99	0.58	3.01
Cortisol (nmol/L)	179	5.49	3.60	0.269	65.57	0.43	17.40
Progesterone (nmol/L)	179	0.76	1.34	0.100	176.66	0	6.77
Erythrocytes (T/L)	182	7.21	0.83	0.062	11.58	5.34	10.03
Hemoglobin (nmol/L)	182	5.87	0.67	0.049	11.39	4.47	7.94
Hematocrit (L/L)	182	0.30	0.04	0.003	12.32	0.22	0.41
MCV (fL)	182	41.47	2.22	0.165	5.35	36.86	47.33
MCHC (nmol/L)	182	19.68	0.58	0.043	2.97	17.51	21.74
MCH (pg)	182	0.82	0.03	0.002	4.24	0.75	0.93
Leukocytes (g/L)	182	13.27	2.22	0.164	16.72	8.60	18.90
Band neutrophils (g/L)	182	0.04	0.06	0.004	174.18	0	0.19
Segmented neutrophils (g/L)	182	3.50	0.95	0.701	27.26	1.62	6.64
Neutrophils (g/L)	182	3.54	0.97	0.702	27.47	1.64	6.80
Eosinophils (g/L)	182	0.66	0.44	0.03	66.76	0.10	2.77
Lymphocytes (g/L)	182	8.67	1.61	0.120	18.63	4.90	13.99
Monocytes (g/L)	182	0.40	0.12	0.009	30.65	0.19	0.76
Platelets (g/L)	182	527.74	182.21	13.507	34.53	190.00	928.00

RT: rectal temperature; LS: superficial temperature of the lower side of the body obtained by infrared thermography; HTI: heat tolerance index, MCV: mean corpuscular volume; MCHC: mean corpuscular hemoglobin concentration; MCH: mean corpuscular hemoglobin.

**Table 5 animals-12-03473-t005:** Least square means the physiological, hormonal, and hematological traits observed in male and female Caracu cattle in the morning and in afternoon periods.

Trait	Period			Sex		
	Morning	Afternoon	*p*-Value	Male	Female	*p*-Value
RT (°C)	38.038 ± 0.044	39.512 ± 0.044	*p* < 0.0001	38.854 ± 0.050	38.696 ± 0.053	*p* = 0.0326
LS (°C)	34.311 ± 0.094	39.434 ± 0.094	*p* < 0.0001	37.083 ± 0.102	36.662 ± 0.108	*p* = 0.0058
Cortisone (nmol/L)	1.344 ± 0.048	1.470 ± 0.048	*p* = 0.0123	1.262 ± 0.057	1.552 ± 0.059	*p* = 0.0007
Cortisol (nmol/L)	5.591 ± 0.338	5.517 ± 0.340	*p* = 0.8775	3.911 ± 0.332	7.197 ± 0.345	*p* < 0.0001
Progesterone (nmol/L)	0.856 ± 0.128	0.698 ± 0.128	*p* = 0.0013	0.219 ± 0.173	1.335 ± 0.183	*p* < 0.0001
Erythrocytes (T/L)	7.341 ± 0.087	7.088 ± 0.087	*p* = 0.0003	7.107 ± 0.110	7.322 ± 0.116	*p* = 0.1818
Hemoglobin (nmol/L)	5.979 ± 0.068	5.784 ± 0.068	*p* = 0.0003	5.730 ± 0.087	6.033 ± 0.092	*p* = 0.0182
Hematocrit (L/L)	0.305 ± 0.004	0.293 ± 0.004	*p* < 0.0001	0.291 ± 0.005	0.308 ± 0.005	*p* = 0.0134
MCV (fL)	41.605 ± 0.225	41.403 ± 0.226	*p* = 0.0158	40.923 ± 0.306	42.085 ± 0.324	*p* = 0.0107
MCHC (nmol/L)	19.605 ± 0.061	19.743 ± 0.061	*p* = 0.0027	19.737 ± 0.078	19.610 ± 0.082	*p* = 0.2672
MCH (pg)	0.815 ± 0.003	0.817 ± 0.003	*p* = 0.1865	0.807 ± 0.005	0.825 ± 0.005	*p* = 0.0125
Leukocytes (g/L)	13.247 ± 0.220	13.372 ± 0.220	*p* = 0.4731	12.535 ± 0.278	14.084 ± 0.293	*p* = 0.0002
Band neutrophils (g/L)	0.031 ± 0.006	0.042 ± 0.006	*p* = 0.2119	0.025 ± 0.007	0.047 ± 0.007	*p* = 0.0310
Segmented neutrophils (g/L)	3.426 ± 0.099	3.589 ± 0.099	*p* = 0.0389	3.379 ± 0.126	3.636 ± 0.133	*p* = 0.1638
Neutrophils (g/L)	3.457 ± 0.101	3.631 ± 0.101	*p* = 0.0332	3.405 ± 0.128	3.684 ± 0.135	*p* = 0.1366
Eosinophils (g/L)	0.656 ± 0.046	0.672 ± 0.046	*p* = 0.6120	0.585 ± 0.059	0.742 ± 0.062	*p* = 0.0710
Lymphocytes (g/L)	8.740 ± 0.161	8.656 ± 0.161	*p* = 0.5320	8.172 ± 0.202	9.224 ± 0.213	*p* = 0.0006
Monocytes (g/L)	0.394 ± 0.013	0.414 ± 0.013	*p* = 0.1969	0.374 ± 0.014	0.434 ± 0.014	*p* = 0.0033
Platelets (g/L)	527.961 ± 18.712	522.686 ± 18.712	*p* = 0.3620	569.333 ± 25.420	481.314 ± 26.857	*p* = 0.0194

RT: rectal temperature; LS: superficial temperature of the lower side of the body obtained by infrared thermography; MCV: mean corpuscular volume; MCHC: mean corpuscular hemoglobin concentration; MCH: mean corpuscular hemoglobin.

## Data Availability

The data presented in this study are available on request from the corresponding author. The data are not publicly available due to privacy or ethical restrictions, and the data that support the findings of this study are available in this article.

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
