# Peer review of "Effect of Thermal Stress on Thermoregulation, Hematological and Hormonal Characteristics of Caracu Beef Cattle"

_animals, 2022, doi:10.3390/ani12243473_

Round 1
Reviewer 1 Report
Comments on the manuscript “Effect of Thermal Stress on Physiological, Hematological and Hormonal Traits of Caracu Beef Cattle” submitted to the Animals
General comments
I appreciate the opportunity to review this manuscript. The manuscript describes an experiment on the effects of exposure to the sun and shade on Caracu cattle, reared under natural conditions, at different times of the day. Comparisons extended to the sex and coat pattern of the animals. Possible physiological alterations were described and discussed.
The topic is relevant for many reasons, among which I highlight two: first, knowing aspects of the thermoregulation physiology of a little-known breed helps in its preservation; second, the climatic changes expected in the coming decades may influence the type of cattle raised in herds around the world under conditions of extreme heat.
In the manuscript, the methodology and the description of the results are strong points. However, there are contradictions, conceptual errors, and phrases that seem more speculative than based on the experiment's findings. Discussion is the weakness of this manuscript, as the authors will be able to read in my comments below.
Title
Line 1: The phenotype is the result of genetic, epigenetic, and environmental interactions. Traits are generally associated with phenotypic traits with less variation influenced by the environment. I suggest using more appropriate terms such as "measurements", "characteristics" or "variations".
The dependent variables investigated are all physiological: thermoregulation physiology, hematology, and endocrine physiology. Therefore, the title is redundant when using the term "physiological". Change it.
Simple Summary and Abstract
I believe that if the suggestions are accepted, there may be a change in the text of the Simple Summary and the Abstract
Introduction
Line 53: It is unnecessary to state that cattle are homoeothermic because it is consolidated knowledge in the sciences.
Line 60: As written, the authors seem to base the argument that there is Group Selection. As is well known, selection takes place on individuals, not on the group.
Line 60: All living things are continually undergoing natural selection. In the case of domestic animals for meat production, another force acts on the phenotypic conformation and on the genotype: artificial selection controlled by human beings. At some level, more or less intense, there is human action on domesticated species. Therefore, it is not correct to state that the Caracu breed is the sole result of natural selection.
Line 84: What does " largest effective" mean?
Line 86: I have doubts that the Caracu breed is well known due to its longevity etc. Insert a reference to support this claim.
Line 101: Please, read the comments on the title.
Methods
Line 104: There is a lack of information about the age of the animals and their health status.
Were the females pregnant?
Are males neutered?
Have the animals been certified for breed purity?
Clarify whether there was water ad libitum for the animals during the experiment.
Line 120: According to the formula above, the term is tbg, not tgb.
Results
Line 254: Table 4, Please check whether it is a standard error or standard deviation.
Line 298: In the sub-heading, the authors mean “hair length, coat color, muzzle color”, rather than “hair and skin color”.
General comments on the Discussion
In many parts of the discussion, the authors argue that comparisons with other bovine breeds and with the species B. indicus are difficult (lines 329, 331, 433, and 574). In some sentences, the authors attribute to methodological differences the impossibility of considering comparisons between bovine breeds. These arguments contradict the discussion strategy in the manuscript because the authors insist on making comparisons (lines 326, 428, 446, 449, 458, 460, 465, 507, and 526). In fact, the purpose of the article in the objectives is not to compare with other bovine breeds. Therefore, the comparison should be restricted to the Caracu herd studied, between morning and afternoon and between sexes. The discussion should be focused on the physiology of cattle, not on comparing breeds. For physiological comparisons of heat stress between the Caracu breed and other bovine breeds, the methodology of the article should be different.
Line 401: It is not described in the methodology that the animals were weighed. Animal weights must be in the results.
Lines 406 to 422: I agree with the authors that the variation in cortisol concentration can have many endogenous and environmental influences. In the present study, the authors tried to compare cortisol levels, but they cannot clearly explain the differences between morning and afternoon and between genders. The authors attribute these differences to the reactivity of the females, but there is no description of the behavioral evaluation of the animals. The authors also attribute the differences to the activity of the hypothalamic-adrenal axis, which is a circular explanation because these structures control variations of cortisol in the blood.
Line 422: I don't understand the expression "immunological patterns against homeostasis”. What do you mean?
Line 486: Delete “very”. It is a wordie.
Line 542: An acute infection has etiological and pathogenic features different from heat stress. Thus, it's not comparable. The mechanisms of heat stress are known. Heat stress physiological mechanism is similar to the inflammatory response. I suggest reading a paper authored by Sejian & Srivastava (2010).
Line 557 to 561: It is unreasonable to argue that the higher concentration of platelets in males is due to the larger body volume because you will also find a larger blood volume. The interpretation that progesterone and estrogen hormones can stimulate hemodilution needs to be further discussed because reference [86] is not a supporting experiment. The authors of that article [86] only suggest that hemodilution is increased in Nelore females, citing an article on the performance of weight gain and blood patterns in Sindi cattle (Silva et al., 2005). Silva et al. (2005) also do not tested the influence of hormones, determining a lower concentration of platelets in Sindi females, but mentioned that estrogens can influence the renin-angiotensin system, increasing blood volume. Therefore, it is necessary for the authors to consider other explanations. For example, testosterone and its metabolites, which are normally found in higher concentrations in males, are potent stimulators of hematopoiesis. Even though the authors did not assess testosterone concentration in the present study, this may be a reasonable explanation.
Line 564: Replace the verb to identify with to detect or to find.
Line 571: The authors do not describe in the methods, nor in the results, the study of the behavior or temperament of the animals. Furthermore, the authors insert a reference that is not related to females of the Caracu breed, but to the Limousin breed.
Line 574: It is not clear which hypotheses the authors are referring to.
Line 578: Please, check my comments on sub-heading in the line 298.
Line 586: “Apud”. I strongly discourage the use of indirect references. Reference [24], for example, does not seem to me to be consistently peer-reviewed in scientific journals.
Conclusions
Line 574: Due to the suggestions and the need for substantial changes in the text, the conclusion should be rewritten.
References
Lines 624 to 853: Ninety-nine references seem excessive in an article that is not a review. I believe that if the suggestions are accepted, there may be a substantial reduction in the number of references.
REFERENCES
Sejian, V., & Srivastava, R. S. (2010). Pineal–adrenal–immune system relationship under thermal stress: effect on physiological, endocrine, and non-specific immune response in goats. Journal of physiology and biochemistry, 66(4), 339-349.
Silva, R.M.N., Souza, B.B., Souza, A.P., Marinho, M.L., Tavares, G.P., and Silva, E.M.N., (2005). Efeito do sexo e da idade sobre os parâmetros fisiológicos e hematológicos de bovinos da raça Sindi no semi-árido. Ciência e Agrotecnologia, 29, 193–199.
Author Response
Dear reviewer,
We appreciate your precious time in reviewing our manuscript and providing valuable comments that led to possible improvements in this version. The authors have carefully considered all the suggestions.
Best regards,
Title
Line 1: The phenotype is the result of genetic, epigenetic, and environmental interactions. Traits are generally associated with phenotypic traits with less variation influenced by the environment. I suggest using more appropriate terms such as "measurements", "characteristics" or "variations".
The dependent variables investigated are all physiological: thermoregulation physiology, hematology, and endocrine physiology. Therefore, the title is redundant when using the term "physiological". Change it.
Authors: Thanks for pointing that out. The title has been modified as requested (L1).
Simple Summary and Abstract
I believe that if the suggestions are accepted, there may be a change in the text of the Simple Summary and the Abstract
Authors: All suggestions were accepted and incorporated in Simple Summary (L17-L28) and Abstract (L29-L43) sessions.
Introduction
Line 53: It is unnecessary to state that cattle are homoeothermic because it is consolidated knowledge in the sciences.
Authors: The sentence has been removed.
Line 60: As written, the authors seem to base the argument that there is Group Selection. As is well known, selection takes place on individuals, not on the group.
Authors: I am sorry about this mistake. The sentence has been rewritten (L58-L59).
Line 60: All living things are continually undergoing natural selection. In the case of domestic animals for meat production, another force acts on the phenotypic conformation and on the genotype: artificial selection controlled by human beings. At some level, more or less intense, there is human action on domesticated species. Therefore, it is not correct to state that the Caracu breed is the sole result of natural selection.
Authors: Thanks for pointing that out. We agree with your comments. The sentence has been removed.
Line 84: What does " largest effective" mean?
Authors: It means that Caracu has the largest herd to production (beef and milk) among the Brazilian Creole breeds. The sentence has been modified (L80-L81).
Line 86: I have doubts that the Caracu breed is well known due to its longevity etc. Insert a reference to support this claim.
Authors: Some references have been included to support this claim (L82-L83).
Line 101: Please, read the comments on the title.
Authors: The sentence has been modified according to the title (L97-L98).
Methods
Line 104: There is a lack of information about the age of the animals and their health status. Were the females pregnant?
Authors: The females used in this study were steers virgin (L105).
Are males neutered?
Authors: The males used in this study were intact (L105).
Have the animals been certified for breed purity?
Authors: All animals were certified for breed purity by ABCCaracu (L107-L108).
Clarify whether there was water ad libitum for the animals during the experiment.
Authors: This information is described in L121.
Line 120: According to the formula above, the term is tbg, not tgb.
Authors: Thanks for pointing that out (L117).
Results
Line 254: Table 4, Please check whether it is a standard error or standard deviation.
Authors: Thanks for pointing that out. I am sorry for this mistake. We included the SD in Table 4. In addition, we adopted the SI-units for all characteristics studied, as requested by the other reviewer (Table 4 and Table 5).
Line 298: In the sub-heading, the authors mean “hair length, coat color, muzzle color”, rather than “hair and skin color”.
Authors: Done. The sub-heading has been modified (L247).
General comments on the Discussion
In many parts of the discussion, the authors argue that comparisons with other bovine breeds and with the species B. indicus are difficult (lines 329, 331, 433, and 574). In some sentences, the authors attribute to methodological differences the impossibility of considering comparisons between bovine breeds. These arguments contradict the discussion strategy in the manuscript because the authors insist on making comparisons (lines 326, 428, 446, 449, 458, 460, 465, 507, and 526). In fact, the purpose of the article in the objectives is not to compare with other bovine breeds. Therefore, the comparison should be restricted to the Caracu herd studied, between morning and afternoon, and between sexes. The discussion should be focused on the physiology of cattle, not on comparing breeds. For physiological comparisons of heat stress between the Caracu breed and other bovine breeds, the methodology of the article should be different.
Authors: Thank you for your suggestions. The Discussion section has been rewritten.
Line 401: It is not described in the methodology that the animals were weighed. Animal weights must be in the results.
Authors: The information has been included in the manuscript (L105-L106).
Lines 406 to 422: I agree with the authors that the variation in cortisol concentration can have many endogenous and environmental influences. In the present study, the authors tried to compare cortisol levels, but they cannot clearly explain the differences between morning and afternoon and between genders. The authors attribute these differences to the reactivity of the females, but there is no description of the behavioral evaluation of the animals. The authors also attribute the differences to the activity of the hypothalamic-adrenal axis, which is a circular explanation because these structures control variations of cortisol in the blood.
Authors: Thank you for your suggestion. The sentence has been rewritten. The reactivity of the females has been clearly observed, but not measured in our study. These animals were previously submitted to a feed efficiency study when we could observe the high reactivity of females during all experiments. In addition, the high reactivity of females when compared to males has been described in the literature (L396-L418).
Line 422: I don't understand the expression "immunological patterns against homeostasis”. What do you mean?
Authors: Thanks for pointing that out. It’s a mistake. The sentence has been modified (L424-L425).
Line 486: Delete “very”. It is a wordie.
Authors: Done (L452).
Line 542: An acute infection has etiological and pathogenic features different from heat stress. Thus, it's not comparable. The mechanisms of heat stress are known. Heat stress physiological mechanism is similar to the inflammatory response. I suggest reading a paper authored by Sejian & Srivastava (2010).
Authors: Thanks for pointing that out. The sentence has been modified (L493-L495).
Line 557 to 561: It is unreasonable to argue that the higher concentration of platelets in males is due to the larger body volume because you will also find a larger blood volume. The interpretation that progesterone and estrogen hormones can stimulate hemodilution needs to be further discussed because reference [86] is not a supporting experiment. The authors of that article [86] only suggest that hemodilution is increased in Nelore females, citing an article on the performance of weight gain and blood patterns in Sindi cattle (Silva et al., 2005). Silva et al. (2005) also do not tested the influence of hormones, determining a lower concentration of platelets in Sindi females, but mentioned that estrogens can influence the renin-angiotensin system, increasing blood volume. Therefore, it is necessary for the authors to consider other explanations. For example, testosterone and its metabolites, which are normally found in higher concentrations in males, are potent stimulators of hematopoiesis. Even though the authors did not assess testosterone concentration in the present study, this may be a reasonable explanation.
Authors: Thanks for your suggestion. The sentence has been modified according to your suggestion (L500-L509).
Line 564: Replace the verb to identify with to detect or to find.
Authors: The sentence has been deleted.
Line 571: The authors do not describe in the methods, nor in the results, the study of the behavior or temperament of the animals. Furthermore, the authors insert a reference that is not related to females of the Caracu breed, but to the Limousin breed.
Authors: Thanks for pointing that out. It’s a mistake. The sentence has been modified (L516-L517).
Line 574: It is not clear which hypotheses the authors are referring to.
Authors: The sentence has been modified (L518-L520).
Line 578: Please, check my comments on sub-heading in the line 298.
Authors: Done. The sub-heading has been modified (L524).
Line 586: “Apud”. I strongly discourage the use of indirect references. Reference [24], for example, does not seem to me to be consistently peer-reviewed in scientific journals.
Authors: Thank you for your suggestion (L532).
Conclusions
Line 574: Due to the suggestions and the need for substantial changes in the text, the conclusion should be rewritten.
Authors: The conclusion has been rewritten as requested (L538-L549).
References
Lines 624 to 853: Ninety-nine references seem excessive in an article that is not a review. I believe that if the suggestions are accepted, there may be a substantial reduction in the number of references.
Authors: The number of references has been reduced.
Reviewer 2 Report
Dear Authors, please use the SI-units for hematological parameters and cortisol and progesteron concentrations in the blood and correct the manuscript!

Author Response
Dear reviewer,
We appreciate your precious time in reviewing our manuscript and providing comments that led to possible improvements in this version. The authors have carefully considered all the suggestions.
We adopted the SI-units for all characteristics studied, as requested.
Best regards,